

# Generalised Onsager algebra in quantum lattice models

**Yuan Miao**

Galileo Galilei Institute for Theoretical Physics, INFN,
Largo Enrico Fermi 2, 50125 Firenze, Italy

yuan.miao@fi.infn.it

## Abstract

The Onsager algebra is one of the cornerstones of exactly solvable models in statistical mechanics. Starting from the generalised Clifford algebra, we demonstrate its relations to the graph Temperley–Lieb algebra, and a generalisation of the Onsager algebra. We present a series of quantum lattice models as representations of the generalised Clifford algebra, possessing the structure of a special type of the generalised Onsager algebra [1]. The integrability of those models is presented, analogous to the free fermionic eight-vertex model. We also mention further extensions of the models and physical properties related to the generalised Onsager algebras, hinting at a general framework that includes families of quantum lattice models possessing the structure of the generalised Onsager algebras.

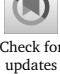

# 1   Introduction

Exactly solvable models [2–4] play an important role in statistical mechanics, providing us with the possibility to obtain analytical and mathematically rigorous results that are scarce when the physical systems are interacting. One of the first examples of exactly solvable models is the Onsager's solution to the two-dimensional classical Ising model [5] in 1944. Onsager used the Onsager algebra to obtain the partition function of the two-dimensional classical Ising model in the absence of the magnetic field. Since then, the Onsager algebra has become a useful tool to study many classical statistical mechanical and quantum lattice models, such as the chiral Potts model [6,7] and the $\mathbb{Z}_N$-symmetric spin chain [8,9]. In addition, the Onsager algebra is closely related to quantum integrability [10] and Kramers–Wannier duality [11–13], offering many facets on understanding exactly solvable models to us. More recently, the Onsager algebra has been conjectured to be present in a series of quantum integrable systems at root of unity values of anisotropy [14–16], e.g. the spin-1/2 quantum XXZ model at root of unity with quasi-periodic boundary conditions [1], hinting at an intriguing relation to the representation theory of quantum groups. The Onsager algebra is also closely related to quantum many-body scars [17], a subset of the eigenstates of non-integrable quantum systems that have non-trivial out-of-equilibrium behaviour. The Onsager algebra can be used to solve the dynamics of interacting quantum systems [18] as well.

From the mathematical perspective, the Onsager algebra is an infinite-dimensional Lie algebra [19], which is a fixed-point subalgebra of the $\mathfrak{sl}_2$ loop algebra [15,20]. Recently the alternating presentation and its central extension of the Onsager algebra has been studied in [21,22]. The close relation between the Onsager algebra and the classical Yang–Baxter algebras has been investigated in [23]. There are few ways to extend the Onsager algebra. One is to consider the $q$-deformation of the model, the $q$-Onsager algebra [24–26], a coideal subalgebra of affine $U_q(\hat{\mathfrak{sl}}_2)$, with connections to the quantum XXZ model with open boundary conditions. The alternating presentation and its central extension of the $q$-Onsager algebra can be found in [27,28]. Meanwhile, one can consider a generalisation of the Onsager algebra as a Lie subalgebra of certain Kac-Moody algebra that satisfies Dolan–Grady-like relations. One of the first attempts is the so-called "$\mathfrak{sl}_n$ Onsager algebra" [29], i.e. a Lie subalgebra of the affine Lie algebra $A_{n-1}^{(1)}$. Results for other generalisations can be found in [30], even in the presence of the $q$-deformation [25,31]. Recently, a systematic construction of this type of generalisations was presented in [1], dubbed "generalised Onsager algebras", classifying all the Lie subalgebras of different Kac–Moody algebras with Dolan–Grady-like relations. In the meantime, the Yang–Baxter algebra presentation of the generalised Onsager algebras has been studied in [23,32]. The result of [1] serves as a motivation for this article, where we try to find physically relevant models that consists of parts as the representation of a certain generalised

---

[1]For certain roots of unity, e.g. $q = \exp(i\pi/2)$ for spin 1/2 or $q = \exp(i\pi/3)$ for spin 1, the Onsager algebra can be found explicitly for the spin-1/2 XXZ model or the spin-1 Zamolodchikov–Fateev model [14], where the Onsager generators consist of local operators. When we are at other roots of unity, the Onsager generators are expected to have quasi-local density [15].

Onsager algebra. Indeed, we discover an elegant connection between the generalised Clifford algebra, defined in Section 2, and one type of the generalised Onsager algebras, defined in [1], which possesses a representation related to the Fendley model [33–36], a model of medium-range spin interactions having free fermionic spectra with open boundary conditions. We show that the Fendley model with periodic boundary condition (i.e. interacting) can be expressed in terms of operators satisfying the generalised Onsager algebra, and is integrable, where we present a different approach compared to the one in [33], analogous to the free fermionic eight-vertex model.

The outline of the article is as follows. We introduce the generalised Clifford algebra, the key figure of this article, and its relation to the graph Temperley-Lieb algebra and generalised Onsager algebra. We present the physically relevant representations of the generalised Clifford algebra, including the transverse field Ising model and free fermionic eight-vertex model. Most importantly, the Fendley model consists of operators that belong to a representation of the generalised Clifford algebra as well as a special type of the generalised Onsager algebra, which is the first quantum lattice model associated with the generalised Onsager algebra to the best of our knowledge. Since the Onsager algebra implies integrability of the model, we proceed with presenting the integrability of the Fendley model motivated by recent works on medium-range quantum integrable models [37,38]. Eventually, we present the chiral-Potts-like generalisation of the Fendley model where the generalised Onsager algebra remains, before ending the article with conclusions and outlook.

## 2 Relations among three algebras

We start with defining the generalised Clifford algebra (GCA) [34,35,39,40], whose representation plays a crucial roles in the following.

The generalised Clifford algebra $GC(r,N)$ is a unital associative algebra over the complex numbers with generators $h_1, h_2, \cdots, h_N$ satisfying

$$
\begin{aligned}
&h_j^2 = \mathrm{id}, \quad h_j h_{j+m} = -h_{j+m} h_j, \quad 1 \le m \le r, \\
&h_j h_{j+n} = h_{j+n} h_j \quad n \ge r+1, \ 1 \le j \le N,
\end{aligned}
\tag{1}
$$

with "periodic boundary conditions" $h_{N+k} \equiv h_k$, $k \ge 1$. When $r = 1$, it becomes the usual Clifford algebra. As illustrated in Fig. 1, for $GC(2,6)$,

$$
h_1 h_5 = -h_5 h_1, \quad h_1 h_4 = h_4 h_1,
\tag{2}
$$

etc.

There are many known representations of the GCA that are relevant to exactly solvable models. The most renowned one is the transverse field Ising model (TFIM) as a representation of $GC(1,N)$. The explicit constructions are given in Section 3.1.

We move on to the *"periodic graph Temperley–Lieb (TL) algebra"* $GTL(\beta, r, N)$ [41]. As demonstrated below, we obtain a quotient of the algebra $GTL(\sqrt{2}, r, N)$ from the GCA. It is defined as the unital associative algebra over the complex numbers with generators $e_1, e_2, \cdots e_N$, satisfying

$$
\begin{aligned}
&e_j^2 = \beta e_j, \quad e_j e_{j\pm m} e_j = e_j, \quad 1 \le m \le r, \\
&e_j e_{j+n} = e_{j+n} e_j \quad n \ge r+1, \ 1 \le j \le N,
\end{aligned}
\tag{3}
$$

with $e_{N+k} \equiv e_k$, $k \ge 1$.

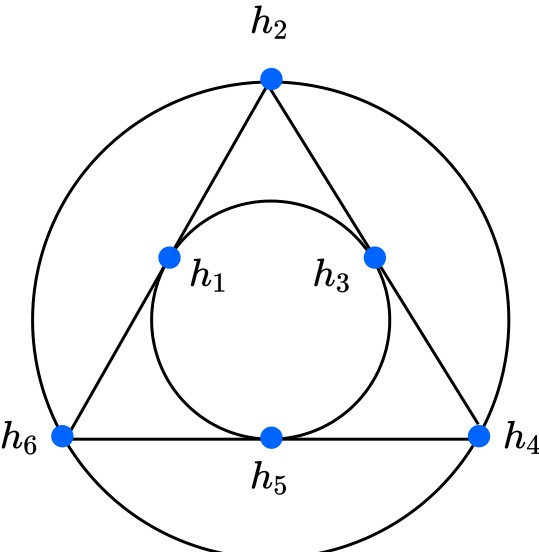

Figure 1: An illustration of the relations between the generators $h_j$ for the algebra GC(2, 6). Each dot corresponds to a generator, and if they are connected via a solid line, they anticommute among each other. If they are not connected, they mutually commute.

**Remark.** The algebra GTL($\beta, 1, N$) is very similar to the periodic (affine) Temperley-Lieb algebra [42], which is a quotient of the affine Hecke algebra [43]. However, the main difference here is that we do not require the existence of the element $g$ from the periodic TL algebra, such that

$$g e_j g^{-1} = e_{j+1}. \tag{4}$$

Usually we can define an operator **G** for the physical relevent representations of GTL($\beta, r, N$) that satisfies the relation (4), in the form of right translational operator. However, it is a bit complicated to define such an operator for Ising or Potts like models. The existence of $g$ does not affect the discussions about the relations to the generalised Onsager algebra. Therefore, we do not require the existence of $g$ in our definition of GTL($\beta, r, N$).

From the algebra GC($r, N$), we can construct a map from a quotient of GTL($\sqrt{2}, r, N$), i.e. GTL$'(\sqrt{2}, r, N)$, to GC($r, N$)

$$e'_j \to \frac{1}{\sqrt{2}}(\mathrm{id} + h_j), \tag{5}$$

satisfying (3).

Generators $e'_j$ satisfy the following relations in addition to the definition of GTL($\sqrt{2}, r, N$) (3),

$$\{e'_j, e'_{j+m}\} = \sqrt{2}(e'_j + e'_{j+m}), \quad 1 \le m \le r. \tag{6}$$

For the sake of convenience, we shall use the algebra GTL($\sqrt{2}, r, N$) and its definition (3) mainly for the rest of the article.

Finally, we focus on a special type of generalisation of the Onsager algebra GO($r + 1$) defined in **Definition 2.5** of [1]. The algebra GO($r + 1$) is an infinite-dimensional Lie algebra. Without diving into the full definition, for which we refer the readers to [1], we use the generalised Dolan–Grady presentation of the algebra GO($r + 1$) with only ($r + 1$)-many generators $A^{(s)}$, $0 \le s \le r$, which are enough to generate the rest of the infinitely many ones. The $r + 1$

generators $A^{(s)}$ satisfy the Dolan–Grady relation between any pair of them,

$$\left[A^{(s)},\left[A^{(s)},\left[A^{(s)},A^{(t)}\right]\right]\right]=16\left[A^{(s)},A^{(t)}\right], \quad \forall s,t \in \{0,1,\cdots,r\}. \tag{7}$$

As shown in [1], the algebra $GO(r+1)$ is a fixed-point Lie subalgebra of a Kac–Moody algebra g with generalised Cartan matrix of dimension $(r+1)$

$$\begin{pmatrix} 2 & -2 & -2 & \cdots & -2 & -2 \\ -2 & 2 & -2 & \cdots & -2 & -2 \\ \vdots & & \ddots & & & \vdots \\ -2 & -2 & -2 & \cdots & 2 & -2 \\ -2 & -2 & -2 & \cdots & -2 & 2 \end{pmatrix}. \tag{8}$$

The mathematical curiosities of the Kac–Moody algebra here are mentioned in Ref. [44], which is not the main focus of this paper.

When $r=1$, we have $GO(2)$, which is the renowned Onsager algebra, which possesses the Kramers–Wannier duality $A^{(0)} \leftrightarrow A^{(1)}$ [12],

$$\begin{aligned} \left[A^{(0)},\left[A^{(0)},\left[A^{(0)},A^{(1)}\right]\right]\right]&=16\left[A^{(0)},A^{(1)}\right], \\ \left[A^{(1)},\left[A^{(1)},\left[A^{(1)},A^{(0)}\right]\right]\right]&=16\left[A^{(1)},A^{(0)}\right]. \end{aligned} \tag{9}$$

The cases of $GO(r+1)$ with $r \geq 2$ therefore possess a generalised version of the Kramers–Wannier duality,

$$A^{(s)} \leftrightarrow A^{(t)}, \quad \forall s,t \in \{0,1,2,\cdots,r\}, \tag{10}$$

that leaves the algebra invariant.

There exists a map $GO(r+1) \to GC(r,N)$ with $N \bmod(r+1)=0$ which is injective, where

$$A^{(s)} \to \sum_{j=1}^{N/(r+1)} h_{(r+1)j-s}, \quad s \in \{0,1,2,\cdots,r\}. \tag{11}$$

Since we also have a map $GTL'(\sqrt{2},r,N) \to GC(r,N)$ (5), we obtain a map $GO(r+1) \to GTL'(\sqrt{2},r,N)$, i.e.

$$A^{(s)} \to \sum_{j=1}^{N/(r+1)} (\sqrt{2}e_{(r+1)j-s} - \mathrm{id}), \quad s \in \{0,1,2,\cdots,r\}. \tag{12}$$

In this section, we have shown that the algebra $GC(r,N)$ has two homomorphisms, one to $GTL'(\sqrt{2},r,N)$ and another to $GO(r+1)$. In the following sections, we will study certain physically relevant representations of the algebra $GC(r,N)$, which constitute the Fendley model [33–36].

## 2.1 Additional commuting operators

The algebra $GC(r,N)$ contains $\frac{(r+1)(r+2)}{2}$ operators that commute with any two of the generators $A^{(s)}$ that do not commute with each other. These operators are referred as "$U(1)$-invariant Hamiltonian" in the case of $GC(1,N)$ in [14,15], when considering the representation related to the transverse field Ising model; we will comment on that in Section 3.1.

Within the algebra $GC(r,N)$ with $N \bmod(r+1) = 0$, the operator that commutes with $A^{(s)}$ and $A^{(t)}$, which are defined in (11), is given as

$$H^{(s,t)} = \frac{\mathrm{i}}{4} \sum_{j=1}^{N/(r+1)} \left( h_{(r+1)j-s} h_{(r+1)j-t} + h_{(r+1)j-t} h_{(r+1)j+r+1-s} \right).$$  (13)

We can prove that

$$\left[ H^{(s,t)}, A^{(s)} \right] = \left[ H^{(s,t)}, A^{(t)} \right] = 0,$$  (14)

using the relation (1) repeatedly, even though $[A^{(s)}, A^{(t)}] \neq 0$. This implies that $H^{(s,t)}$ commutes with all the operators that are generated by $A^{(s)}$ and $A^{(t)}$.

For example, if we consider the case with $r = 1$, i.e. the Onsager algebra, we have

$$H^{(0,1)} = \frac{\mathrm{i}}{2} \sum_{j=1}^{N} h_j h_{j+1},$$  (15)

which commutes with all the infinitely many Onsager generators $A_n$ according to (A.1).

## 3  Representations of $GC(r,N)$

Before we concentrate on the case of the Fendley model, we first review a few well-known representations of the algebra $GC(r,N)$. Many of the examples, such as transverse field Ising model and free fermionic eight-vertex models, have been studied previously. Even so, it is useful to present those examples in terms of the representations of the algebras introduced in Sec. 2, which helps us understand better the Fendley model.

### 3.1  Transverse field Ising model

We start with the representation of $GC(1, 2L)$ which is the transverse field Ising model (TFIM). In this case, we have the map: $GC(1, 2L) \rightarrow \mathrm{End}((\mathbb{C}^2)^{\otimes L})$,

$$h_{2j-1} \mapsto \mathbf{h}_{2j-1}^{\mathrm{IM}} = \sigma_j^z, \quad h_{2j} \mapsto \mathbf{h}_{2j}^{\mathrm{IM}} = \sigma_j^x \sigma_{j+1}^x,$$  (16)

where $\sigma^\alpha$ are the Pauli matrices and $\sigma_j^\alpha = \mathbb{1}_2^{\otimes(j-1)} \otimes \sigma^\alpha \otimes \mathbb{1}_2^{\otimes(L-j)}$ are the local spin-1/2 operators (with $\mathbb{1}_2 = \mathrm{diag}(1,1)$ and $\mathbb{1} = \mathbb{1}_2^{\otimes L}$). Here the periodic boundary condition reads $\sigma_{L+1}^\alpha \equiv \sigma_1^\alpha$.

It is straightforward to check that $\mathbf{h}_j^{\mathrm{IM}}$ satisfy the relations of $GC(1, 2L)$ (1). We can define the quantum Hamiltonian of the TFIM as

$$\mathbf{H}^{\mathrm{IM}}(\lambda) = \sum_{j=1}^{L} \left( \lambda \mathbf{h}_{2j-1}^{\mathrm{IM}} + \mathbf{h}_{2j}^{\mathrm{IM}} \right) = \sum_{j=1}^{L} \left( \lambda \sigma_j^z + \sigma_j^x \sigma_{j+1}^x \right).$$  (17)

We can find representations for $GTL(\sqrt{2}, 1, 2L)$ and $GO(2)$, which are the affine TL algebra and the Onsager algebra, respectively, in the case of the TFIM using the homomorphisms in Section 2. Specifically,

$$e_j \mapsto \mathbf{e}_j^{\mathrm{IM}} = \frac{1}{\sqrt{2}} \left( \mathbb{1} + \mathbf{h}_j^{\mathrm{IM}} \right),$$  (18)

$$A^{(0)} \mapsto \mathbf{A}_{\mathrm{IM}}^{(0)} = \sum_{j=1}^{L} \mathbf{h}_{2j-1}^{\mathrm{IM}} = \sum_{j=1}^{L} \sigma_j^z, \quad A^{(1)} \mapsto \mathbf{A}_{\mathrm{IM}}^{(1)} = \sum_{j=1}^{L} \mathbf{h}_{2j}^{\mathrm{IM}} = \sum_{j=1}^{L} \sigma_j^x \sigma_{j+1}^x.$$  (19)

The same construction of (16) and its relation to the Onsager algebra have been studied in [45, 46].

The operators $\mathbf{A}_{\mathrm{IM}}^{(0)}$ and $\mathbf{A}_{\mathrm{IM}}^{(1)}$ satisfy the Dolan–Grady relation (9) and the duality between the two operators is the renowned Kramers–Wannier duality. In fact, we can rewrite the TFIM Hamiltonian in terms of the Onsager generators,

$$\mathbf{H}^{\mathrm{IM}}(\lambda) = \lambda \mathbf{A}_{\mathrm{IM}}^{(0)} + \mathbf{A}_{\mathrm{IM}}^{(1)}, \tag{20}$$

and by using the self-duality we can predict a phase transition happening at $\lambda \to 1$ [12, 47].

The TFIM is solvable, as it can be mapped into a free fermionic model via the Jordan–Wigner transformation, and it can be considered as the quantum limit of the two-dimensional classical Ising model [47–49]. The relation between the algebra GO(1, 2$L$) and generalisations of the Jordan–Wigner transformation has been discussed in [45, 46, 50]. A different generalisation of to the cluster XY-models that possess a similar structure of the Onsager algebra is presented in [51].

The additional conserved operator here is of particular interest. In the case of TFIM, we have

$$\mathbf{H}_{\mathrm{IM}}^{(0,1)} = \frac{\mathrm{i}}{2} \sum_{j=1}^{2L} \mathbf{h}_j^{\mathrm{IM}} \mathbf{h}_{j+1}^{\mathrm{IM}} = \sum_{j=1}^{L} \frac{1}{2} \left( \sigma_j^x \sigma_{j+1}^y - \sigma_j^y \sigma_{j+1}^x \right), \tag{21}$$

which is the spin current of the spin-1/2 XXZ model [52]. With a unitary transformation, $\mathbf{H}_{\mathrm{IM}}^{(0,1)}$ becomes the spin-1/2 XX model, i.e. spin-1/2 XXZ model at root of unity $q = \mathrm{i}$. This means that spin-1/2 XX model, despite being free fermionic, commutes with all the Onsager generators, cf. Appendix A. This observation serves as a starting point for the conjecture of the presence of the Onsager algebra symmetry for the spin-1/2 XXZ model at arbitrary root of unity [15, 16].

In addition to the well-known TFIM, there are a few more physically relevant representations of the algebra GC($r$, $N$) as we present in the following.

## 3.2 Free fermionic eight-vertex models

Similar to the TFIM, there exists another free fermionic representation of the algebra GC(1, $L$). In this case, we again take into account the vector space $(\mathbb{C}^2)^{\otimes L}$ with the following map: GC(1, $L$) → End($(\mathbb{C}^2)^{\otimes L}$)

$$h_j \mapsto \mathbf{h}_{j,j+1}^{\mathrm{8V}} = \sigma_j^\alpha \sigma_{j+1}^\beta =: \mathbf{h}_j^{\mathrm{8V}}, \tag{22}$$

where we pick any $\alpha \neq \beta$ with $\alpha, \beta \in \{x, y, z\}$. Different choices of $\sigma_j^\alpha \sigma_{j+1}^\beta$ are related by a unitary transformation. Without loss of generality, we focus on the case

$$\mathbf{h}_j^{\mathrm{8V}} = \sigma_j^y \sigma_{j+1}^x. \tag{23}$$

We consider the following Hamiltonian,

$$\mathbf{H}^{\mathrm{8V}} = \sum_{j=1}^{L} \mathbf{h}_j^{\mathrm{8V}} = \sum_{j=1}^{L} \sigma_j^y \sigma_{j+1}^x, \tag{24}$$

which we shall call (a special case of) the free fermionic eight-vertex model. The reason why this model is related to the free fermionic eight-vertex model is given in Section 4.1, when studying the integrability of the model.

We can get a representation of the affine TL algebra and the Onsager algebra by considering the same construction as before, i.e.

$$e_j \mapsto \mathbf{e}_j^{\mathrm{8V}} = \frac{1}{\sqrt{2}} \left( \mathbb{1} + \mathbf{h}_j^{\mathrm{8V}} \right), \tag{25}$$

$$A^{(0)} \mapsto \mathbf{A}^{(0)}_{8V} = \sum_{j=1}^{L/2} \mathbf{h}^{8V}_{2j-1} = \sum_{j=1}^{L/2} \sigma^y_{2j-1} \sigma^x_{2j} \,,$$

$$A^{(1)} \mapsto \mathbf{A}^{(1)}_{8V} = \sum_{j=1}^{L/2} \mathbf{h}^{8V}_{2j} = \sum_{j=1}^{L/2} \sigma^y_{2j} \sigma^x_{2j+1} \,,$$

(26)

if the system size $L \bmod 2 = 0$. When we consider the corresponding inhomogeneous Hamiltonian (essentially the coupling constants are staggered),

$$\tilde{\mathbf{H}}^{8V}(\lambda) = \lambda \mathbf{A}^{(0)}_{8V} + \mathbf{A}^{(1)}_{8V} \,, \quad \tilde{\mathbf{H}}^{8V}(1) = \mathbf{H}^{8V} \,,$$

(27)

we can predict the phase transition at the homogenous limit $\lambda \to 1$, again using the self-duality.

The model we consider here can be solved using free fermions by conducting the Jordan–Wigner transformation too, which we will not expand on the details here. Moreover, this simple model can be further generalised into more intriguing cases, which are intrinsically interacting and are related to the generalised version for the affine TL and Onsager algebras.

## 3.3 Fendley model

With the example of the free fermionic eight-vertex model, we can generalise to the representation of $\mathrm{GC}(r, L)$ with $r \geq 2$. We consider the map: $\mathrm{GC}(r, L) \to \mathrm{End}((\mathbb{C}^2)^{\otimes L})$,

$$h_j \mapsto \mathbf{h}^{\mathrm{FM}}_{j,j+1,\cdots,j+r} = \sigma^\alpha_j \sigma^\beta_{j+1} \cdots \sigma^\beta_{j+r} =: \mathbf{h}^{\mathrm{FM}}_{[j,j+r]} \,.$$

(28)

The following Hamiltonian

$$\mathbf{H}^{\mathrm{FM}}(r, \{\xi_m\}^L_{m=1}) = \sum_{j=1}^{L} \xi_m \mathbf{h}^{\mathrm{FM}}_{[j,j+r]} \,,$$

(29)

which has first been considered in [33] for the $r = 2$ cases and subsequently in [34,35] for the more general cases. Hence, we refer to the model as the Fendley model. The Hamiltonians with the open boundary condition, i.e.

$$\mathbf{H}^{\mathrm{FM}}_{\mathrm{open}}(r, \{\xi_m\}^L_{m=1}) = \sum_{j=1}^{L-r} \xi_m \mathbf{h}^{\mathrm{FM}}_{[j,j+r]}$$

(30)

has free fermionic spectra [2] which can be diagonalised using a non-local transformation into fermionic bilinears. However, the spectra of the Hamiltonians do not satisfy the free fermionic condition in [53] with the periodic boundary condition. This can be observed from numerically obtaining the eigenvalues of (29), as shown in Fig. 2. In principle, this do not exclude the possibility that the spectra cannot be partitioned into subsectors that are free fermionic. The most notable example is the TFIM with periodic boundary condition, where the spectrum can be divided into two parts that are free fermionic. In the case of the Fendley model, it is less clear whether such partition of the spectrum into free fermionic parts exists. Moreover, unlike the TFIM, the non-local transformation constructed in [33] no longer applies for periodic boundary. The question whether the Fendley model is intrinsically interacting is postponed to future investigation.

---

[2]What we refer to as "free fermionic spectra" is defined in Eq. (1) of [53].



Figure 2: The energy spectrum of the homogeneous Fendley model (31) with system size $L = 6$. The red numbers above are the degeneracies of the different energy eigenvalues.

If we choose the coupling to be homogeneous $\xi_m = 1$, i.e.

$$\mathbf{H}^{\text{FM}}(r) := \mathbf{H}^{\text{FM}}(r, \xi_m = 1) = \sum_{j=1}^{L} \mathbf{h}^{\text{FM}}_{[j,j+r]}, \tag{31}$$

the model is integrable, whose transfer matrices with a three-dimensional auxiliary space have been constructed in [33] for the case of $r = 2$. In this article, we instead take a different approach similar to [37] which has a four-dimensional auxiliary space (tensored space with two spin-1/2s) when $r = 2$, revealing a hidden connection to the integrability of the free fermionic eight-vertex model, presented in Sec. 4.2.

**Remark.** The transfer matrix in [33] can be applied to the $r = 2$ Fendley model with inhomogenous couplings and periodic boundary. However, the method in Sec. 4.2 only works for the homogenous case (31). Instead, we can add inhomogeneities in the transfer matrix (64) constructed in Sec. 4.2, which will result in another Hamiltonian with longer-range interaction.

There is a hidden supersymmetric algebra too, desribed in [33]. The observation is that there is a "dual representation" for the same algebra GC($L, r$),

$$h_j \mapsto \tilde{\mathbf{h}}^{\text{FM}}_{[j,j+r]} = \sigma^{\beta}_j \sigma^{\beta}_{j+1} \cdots \sigma^{\beta}_{j+r-1} \sigma^{\alpha}_{j+r}, \tag{32}$$

such that

$$\left[ \mathbf{h}^{\text{FM}}_{[j,j+r]}, \tilde{\mathbf{h}}^{\text{FM}}_{[k,k+r]} \right] = 0, \quad \forall j, k \in \{1, 2, \cdots, L\}. \tag{33}$$

These two representations (in the case of the periodic boundary condition) are related via a Clifford transformation [54], i.e.

$$\text{CT}: \quad \sigma^{\alpha}_j \mapsto \sigma^{\beta}_{j-r} \cdots \sigma^{\beta}_{j-1} \sigma^{\alpha}_j \sigma^{\beta}_{j+1} \cdots \sigma^{\beta}_{j+r}, \quad \sigma^{\beta}_j \mapsto \sigma^{\beta}_j, \tag{34}$$

such that

$$\text{CT}: \quad \mathbf{h}^{\text{FM}}_{[j,j+r]} \mapsto \tilde{\mathbf{h}}^{\text{FM}}_{[j-r,j]}. \tag{35}$$

The algebraic relations between the two "dual representations" can be found in [33].

This implies that the "dual Hamiltonian"

$$\tilde{\mathbf{H}}^{\text{FM}}(r) = \sum_{j=1}^{L} \tilde{\mathbf{h}}^{\text{FM}}_{[j,j+r]} \tag{36}$$

commutes with $\mathbf{H}^{\text{FM}}(r)$. This relation motivates us to consider a more general Hamiltonian

$$\mathbf{H}^{\text{FM}}(r, \theta) = \cos\theta \, \mathbf{H}^{\text{FM}}(r) + \sin\theta \, \tilde{\mathbf{H}}^{\text{FM}}(r), \quad 0 \le \theta < 2\pi, \tag{37}$$

which are conjectured to be integrable too for arbitrary $r \in \mathbb{Z}_{>0}$ and $0 \le \theta < 2\pi$ in Sec. 4.2.

We can get a representation of the algebra $\text{GTL}(\sqrt{2}, r, L)$ and the generalised Onsager algebra $\text{GO}(r+1)$ by considering the same construction as before, i.e.

$$e_j \mapsto \mathbf{e}^{\text{FM}}_{[j,j+r]} = \frac{1}{\sqrt{2}} \left( \mathbb{1} + \mathbf{h}^{\text{FM}}_{[j,j+r]} \right), \tag{38}$$

$$A^{(s)} \mapsto \mathbf{A}^{(s)}_{\text{FM}} = \sum_{j=1}^{L/(r+1)} \mathbf{h}^{\text{FM}}_{[(r+1)j+s,(r+1)j+s+r]} = \sum_{j=1}^{L/(r+1)} \sigma^{\alpha}_{(r+1)j+s} \sigma^{\beta}_{(r+1)j+s+1} \cdots \sigma^{\beta}_{(r+1)(j+1)+s}, \tag{39}$$

if the system size $L \bmod (r+1) = 0$. The generalised Onsager algebra hints at the existence of phase transitions in the inhomogeneous Fendley model with inhomogeneous couplings varying with period $r+1$, i.e.

$$\mathbf{H}^{\text{FM}}_{\text{inhomo}}(r, \{\xi_m\}^r_{m=0}) \sum_{j=1}^{L/(r+1)} \sum_{s=0}^{r} \xi_s \mathbf{h}^{\text{FM}}_{[(r+1)j+s,(r+1)j+s+r]}. \tag{40}$$

Taking into account the Kramers–Wannier duality between any two of the generators $\mathbf{A}^{(s)}_{\text{FM}}$, we expect a phase transition when

$$\xi_s = \xi_t; \quad \xi_m < \xi_s, m \neq s, m \neq t, \quad \xi_n > 0, \, 0 \leq n \leq r, \tag{41}$$

which is analogous to the phase transition between the ordered and disordered phases in the TFIM. The case with $r = 2$ has been demonstrated using different method in [33], cf. Fig. 1 in [33].

Moreover, without losing generality, we choose the representation to be

$$\sigma^{\alpha} \equiv \sigma^y, \quad \sigma^{\beta} \equiv \sigma^x, \tag{42}$$

while the other different choices can be obtained via a unitary transformation. This specific choice makes the expressions of the R matrices in the integrability part much easier to visualise, as shown in Sec. 4.2.

## 4 Integrability

As we have recalled in the previous sections, there are several physically relevant representations of the algebra $\text{GC}(L, r)$, which also imply the existence of representations for algebras $\text{GTL}(\sqrt{2}, r, L)$ and $\text{GO}(r+1)$. When $r = 1$, we know that Temperley-Lieb and Onsager algebras indicate the integrability of certain representations as physical Hamiltonian, such as the transverse field Ising model, the chiral Potts model and the spin-1/2 XXZ model at root of unity as recently conjectured in [15].

In the following section, we shall start with an overview of how integrability works for the representation of the free fermionic eight-vertex models with $r = 1$. Then we proceed with the integrability of the Fendley model, focusing on the case of $r = 2$, where an intriguing observation between the integrable structures of the free fermionic eight-vertex models and of the Fendley models are made. We also present a conjecture for the generic integer values of $r$, where the R matrices are more complicated to construct explicitly.

### 4.1 Integrability of the free fermionic eight-vertex model

First of all, we consider the representation in (22) with the Hamiltonian in (24). We can write down the Lax operator and R matrix as

$$\mathbf{R}_{a,j}^{8V}(u) = \mathbf{L}_{a,j}^{8V}(u) = \left( \mathbb{1} + \frac{\sinh(u)}{\sinh(u + i\pi/2)} \mathbf{h}_{a,j}^{8V} \right) \mathbf{P}_{a,j}, \tag{43}$$

where the permutation operator

$$\mathbf{P}_{a,b} = \frac{1}{2} \left( \mathbb{1} + \sum_{\alpha = x,y,z} \sigma_a^\alpha \sigma_b^\alpha \right), \tag{44}$$

such that $\mathbf{P}_{a,b}\mathbf{O}_a = \mathbf{O}_b\mathbf{P}_{a,b}$. This construction of the R matrix can be considered as a reminiscence of the baxterization for the braid-monoid algebra [42, 55–57], i.e.

$$\mathbf{R}(x) \sim (\mathbb{1} + x\mathbf{e})\mathbf{P}, \tag{45}$$

where $\mathbf{e}$ is a representation of the (affine) Temperley-Lieb algebra and $x$ is a certain parametrisation of the spectral parameter, which might not guarantee the R matrix is of difference form [3] in this parametrisation. In principle, the parametrisation of the Yang-Baxter relation for the free fermionic eight-vertex model should be of elliptic type [6, 58]. In our case, we consider a specific case of the free fermionic eight-vertex model, where the trigonometric parametrisation is sufficient.

The R matrix satisfies the Yang–Baxter relation of difference form, i.e. $\mathbf{R}^{8V}(u,v) = \mathbf{R}^{8V}(u-v)$,

$$\mathbf{R}_{a,b}^{8V}(u-v)\mathbf{R}_{a,c}^{8V}(u)\mathbf{R}_{b,c}^{8V}(v) = \mathbf{R}_{b,c}^{8V}(v)\mathbf{R}_{a,c}^{8V}(u)\mathbf{R}_{a,b}^{8V}(u-v). \tag{46}$$

Therefore, we can define the monodromy matrix and transfer matrix as

$$\mathbf{M}_a^{8V}(u) = \prod_{j=1}^{L} \mathbf{L}_{a,j}^{8V}(u), \quad \mathbf{T}^{8V}(u) = \mathrm{tr}_a \mathbf{M}_a^{8V}(u). \tag{47}$$

From the Yang–Baxter relation, the transfer matrices are in involution,

$$\left[ \mathbf{T}^{8V}(u), \mathbf{T}^{8V}(v) \right] = 0, \quad \forall u, v \in \mathbb{C}, \tag{48}$$

indicating the existence of local conserved charges

$$\mathbf{Q}_{n+1}^{8V} = \mathrm{i}\partial_u^n \log \mathbf{T}^{8V}(u)\big|_{u=0}, \tag{49}$$

where $\mathbf{Q}_2^{8V} = \mathbf{H}^{8V}$ is the Hamiltonian.

If we write down the R matrix explicitly, we have

$$\mathbf{R}_{m,n}^{8V}(u) = \begin{pmatrix} a_1 & 0 & 0 & d_1 \\ 0 & b_1 & c_1 & 0 \\ 0 & c_2 & b_2 & 0 \\ d_2 & 0 & 0 & a_2 \end{pmatrix}_{m,n} = \begin{pmatrix} 1 & 0 & 0 & -\tanh u \\ 0 & -\tanh u & 1 & 0 \\ 0 & 1 & \tanh u & 0 \\ \tanh u & 0 & 0 & 1 \end{pmatrix}_{m,n}, \tag{50}$$

which is a special case of the renowned eight-vertex model with $a_1 = a_2 = c_1 = c_2 = 1$, $b_1 = -b_2 = d_1 = -d_2 = \tanh u$. Moreover, the R matrix satisfies the free-fermion condition [2],

$$a_1 a_2 + b_1 b_2 = c_1 c_2 + d_1 d_2, \tag{51}$$

---

[3]The statement of "the R matrix is of difference form" means that $\mathbf{R}(u,v) = \mathbf{R}(u-v)$.

hence a special case of the free fermionic eight-vertex models.

In fact, we can generalise the construction of the integrable structure above to a more generic free fermionic eight-vertex model. Consider a Hamiltonian as the combination of $\mathbf{h}_j^{8V}$ and its dual $\tilde{\mathbf{h}}_j^{8V} = \sigma_j^x \sigma_{j+1}^y$ [4], i.e.

$$H^{8V}(\theta) = \sum_{j=1}^{L} h_j^{8V}(\theta) = \sum_{j=1}^{L} \cos\theta \, \mathbf{h}_j^{8V} + \sin\theta \, \tilde{\mathbf{h}}_j^{8V}. \tag{52}$$

The model again can be mapped into a free fermionic one using Jordan-Wigner transformation. However, the construction of the Lax operator would be useful when considering a similar scenario for the Fendley model. For the new Hamiltonian, we propose the following Lax operator

$$L_{a,j}^{8V}(z,\theta) = \left[ \mathbb{1} + \frac{\sqrt{2}}{2}\left(\frac{z - z^{-1}}{2}\right) h_{a,j}^{8V}(\theta) + \frac{1}{2}\left(\frac{z + z^{-1}}{2} - 1\right)\left(h_{a,j}^{8V}(\theta)\right)^2 \right] \mathbf{P}_{a,j}, \tag{53}$$

where there exists a R matrix that serves as the intertwiner for the Yang–Baxter relation,

$$R_{a,b}^{8V}(z,w,\theta) L_{a,j}^{8V}(z,\theta) L_{b,j}^{8V}(w,\theta) = L_{b,j}^{8V}(w,\theta) L_{a,j}^{8V}(z,\theta) R_{a,b}^{8V}(z,w,\theta). \tag{54}$$

The main difference is that now the R matrix is not possible to be brought into a difference form for $\theta \neq n\pi/2$, $n \in \mathbb{Z}$. The explicit expression for the R matrix can be found in Appendix B.

The Lax operator can be cast into

$$L_{m,n}^{8V}(z,\theta) = \begin{pmatrix} a_1' & 0 & 0 & d_1' \\ 0 & b_1' & c_1' & 0 \\ 0 & c_2' & b_2' & 0 \\ d_2' & 0 & 0 & a_2' \end{pmatrix}_{m,n}, \tag{55}$$

where the coefficients are

$$
\begin{aligned}
a_1' &= a_2' = \frac{1}{4z}\left[(z+1)^2 + (z-1)^2 \sin(2\theta)\right], \\
b_1' &= -b_2' = \frac{1}{2\sqrt{2}iz}(z^2 - 1)(\cos\theta - \sin\theta), \\
c_1' &= c_2' = \frac{1}{4z}\left[(z+1)^2 + (z-1)^2 \sin(2\theta)\right], \\
d_1' &= d_2' = \frac{1}{2\sqrt{2}iz}(z^2 - 1)(\cos\theta + \sin\theta).
\end{aligned}
\tag{56}
$$

Here the free fermion condition is satisfied as well

$$a_1' a_2' + b_1' b_2' = c_1' c_2' + d_1' d_2'. \tag{57}$$

Naturally we can construct the monodromy matrix and the transfer matrix such that

$$M_a^{8V}(z,\theta) = \prod_{j=1}^{L} L_{a,j}^{8V}(z,\theta), \quad T^{8V}(z,\theta) = \mathrm{tr}_a M_a^{8V}(z,\theta), \tag{58}$$

such that

$$\left[T^{8V}(z,\theta), T^{8V}(w,\theta)\right] = 0, \quad z, w \in \mathbb{C}, \tag{59}$$

with the Hamiltonian (52) as a local conserved charge from the transfer matrix

$$H^{8V} = i\partial_z \log T^{8V}(z,\theta)\big|_{z=1}. \tag{60}$$

---

[4]They satisfy the same relations as (33).

## 4.2 Generalisation to $r = 2$ Fendley model

Similar to the free fermionic eight-vertex model, we construct the Lax operator for the Fendley model with $r = 2$, cf. (31). Since we would like the Hamiltonian to be the second conserved charges beside momentum, which has local density acting on three consecutive sites, it is natural to consider a four-dimensional auxiliary space (i.e. 2 two-dimensional auxiliary spaces $a$ and $b$) following the construction in [37], i.e.

$$\mathbf{L}^{\mathrm{FM}}_{a,b,j}(u) = \left( \mathbb{1} + \frac{\sinh(u)}{\sinh(u + \mathrm{i}\pi/2)} \mathbf{h}^{\mathrm{FM}}_{a,b,j} \right) \mathbf{P}_{b,j} \mathbf{P}_{a,j} \,, \tag{61}$$

with the local terms of Hamiltonian being $\mathbf{h}^{\mathrm{FM}}_{a,b,j} = \sigma^y_a \sigma^x_b \sigma^x_j$.

Explicitly, we can express the Lax operator using the coefficients of the R matrix of the free fermionic eight-vertex model (50), i.e.

$$\mathbf{L}^{\mathrm{FM}}_{a,b,j}(u) = \begin{pmatrix} 1 & 0 & 0 & 0 & 0 & 0 & 0 & -\tanh u \\ 0 & 0 & 1 & 0 & 0 & -\tanh u & 0 & 0 \\ 0 & 0 & 0 & -\tanh u & 1 & 0 & 0 & 0 \\ 0 & -\tanh u & 0 & 0 & 0 & 0 & 1 & 0 \\ 0 & 1 & 0 & 0 & 0 & 0 & \tanh u & 0 \\ 0 & 0 & 0 & 1 & \tanh u & 0 & 0 & 0 \\ 0 & 0 & \tanh u & 0 & 0 & 1 & 0 & 0 \\ \tanh u & 0 & 0 & 0 & 0 & 0 & 0 & 1 \end{pmatrix}_{a,b,j} . \tag{62}$$

Indeed, with the parametrisation of the Lax operator (61), there exists the R matrix that serves as the intertwiner for the Yang–Baxter relation,

$$\mathbf{R}^{\mathrm{FM}}_{(a,b)(c,d)}(u,v)\mathbf{L}^{\mathrm{FM}}_{a,b,j}(u)\mathbf{L}^{\mathrm{FM}}_{c,d,j}(v) = \mathbf{L}^{\mathrm{FM}}_{c,d,j}(v)\mathbf{L}^{\mathrm{FM}}_{a,b,j}(u)\mathbf{R}^{\mathrm{FM}}_{(a,b)(c,d)}(u,v) \,, \tag{63}$$

where the R matrix acts non-trivially on 2 four-dimensional auxiliary spaces, i.e. a $16 \times 16$ matrix. The R matrix is not of difference form, and it can be obtained via solving linear equations from the Yang–Baxter relation. The result of the R matrix is presented in Appendix C.

With the Yang–Baxter relation, we construct the monodromy matrix and the transfer matrix by tracing over both auxiliary spaces $a$ and $b$,

$$\mathbf{M}^{\mathrm{FM}}_{a,b}(u) = \prod_{j=1}^{L} \mathbf{L}_{a,b,j}(u) \,, \quad \mathbf{T}^{\mathrm{FM}}(u) = \mathrm{tr}_{a,b} \mathbf{M}^{\mathrm{FM}}_{a,b}(u) \,, \tag{64}$$

such that

$$\left[ \mathbf{T}^{\mathrm{FM}}(u), \mathbf{T}^{\mathrm{FM}}(v) \right] = 0 \,, \quad \forall u, v \in \mathbb{C} \,. \tag{65}$$

We can therefore construct conserved charges with local density by taking the logarithmic derivative of the transfer matrix, the same as in the free fermionic eight-vertex models,

$$\mathbf{Q}^{\mathrm{FM}}_{n+1} = \mathrm{i}\partial^n_u \log \mathbf{T}^{\mathrm{FM}}(u)\big|_{u=0} \,, \tag{66}$$

in particular, the Hamiltonian can be written as

$$\mathbf{H}^{\mathrm{FM}} = \mathbf{Q}^{\mathrm{FM}}_2 = \mathrm{i}\partial^1_u \log \mathbf{T}^{\mathrm{FM}}(u)\big|_{u=0} \,. \tag{67}$$

More interestingly, the density of the higher order charges can be expressed as the Hamiltonian's local terms, e.g.

$$\begin{aligned} \mathbf{Q}^{\mathrm{FM}}_3 &= \mathrm{i}\partial^2_u \log \mathbf{T}^{\mathrm{FM}}(u)\big|_{u=0} \\ &= -2\mathrm{i}\sum_{j=1}^{L} \left( \mathbf{h}^{\mathrm{FM}}_j \mathbf{h}^{\mathrm{FM}}_{j+1} + \mathbf{h}^{\mathrm{FM}}_j \mathbf{h}^{\mathrm{FM}}_{j+2} \right) + L \,. \end{aligned} \tag{68}$$

When the system size $L \bmod 3 = 0$, we define the additional commuting operators for the generalised Onsager generators the same as (13), i.e.

$$\mathbf{H}_{\mathrm{FM}}^{(s,t)} = \frac{\mathrm{i}}{2} \sum_{j=1}^{L/3} \mathbf{h}_{3j-s}^{\mathrm{FM}} \mathbf{h}_{3j-t}^{\mathrm{FM}} + \mathbf{h}_{3j-t}^{\mathrm{FM}} \mathbf{h}_{3j+3-s}^{\mathrm{FM}}. \tag{69}$$

In this case, we express the third-order charge as the sum of the additional commuting operators (while they do not commute among themselves),

$$\mathbf{Q}_3^{\mathrm{FM}} = -4(\mathbf{H}_{\mathrm{FM}}^{(0,1)} + \mathbf{H}_{\mathrm{FM}}^{(0,2)} + \mathbf{H}_{\mathrm{FM}}^{(1,2)}) + L. \tag{70}$$

Similar to the free fermionic eight-vertex model, the "dual Hamiltonian" for the Fendley model reads

$$\tilde{H}^{\mathrm{FM}} = \sum_{j=1}^{L} \tilde{\mathbf{h}}_{[j,j+r]}^{\mathrm{FM}}, \quad \tilde{\mathbf{h}}_{[j,j+r]}^{\mathrm{FM}} = \sigma_j^x \sigma_{j+1}^x \cdots \sigma_{j+r-1}^x \sigma_{j+r}^y, \tag{71}$$

with $\left[ \tilde{\mathbf{h}}_{[j,j+r]}^{\mathrm{FM}}, \mathbf{h}_{[k,k+r]}^{\mathrm{FM}} \right] = 0, \forall j, k$.

When $r = 2$, we have $\tilde{\mathbf{h}}_{[j,j+2]} = \sigma_j^x \sigma_{j+1}^x \sigma_{j+2}^y$. We thus conjecture that the total Hamiltonian

$$\mathsf{H}^{\mathrm{FM}}(\theta) = \sum_{j=1}^{L} \mathsf{h}_j^{\mathrm{FM}}(\theta), \quad \mathsf{h}_j^{\mathrm{FM}}(\theta) = \cos \theta \mathbf{h}_j^{\mathrm{FM}} + \sin \theta \tilde{\mathbf{h}}_j^{\mathrm{FM}} \tag{72}$$

is integrable too. In fact, imitating the construction of (53), we conjecture the following Lax operator

$$\mathsf{L}_{a,b,j}^{\mathrm{FM}}(z,\theta) = \left[ \mathbb{1} + \frac{\sqrt{2}}{2} \left( \frac{z - z^{-1}}{2} \right) \mathsf{h}_{a,b,j}^{\mathrm{FM}}(\theta) + \frac{1}{2} \left( \frac{z + z^{-1}}{2} - 1 \right) \left( \mathsf{h}_{a,b,j}^{\mathrm{FM}}(\theta) \right)^2 \right] \mathbf{P}_{b,j} \mathbf{P}_{a,j} \tag{73}$$

gives rise the monodromy matrix and the transfer matrix defined as

$$\mathsf{M}_{a,b}^{\mathrm{FM}}(z,\theta) = \prod_{j=1}^{L} \mathsf{L}_{a,b,j}^{\mathrm{FM}}(z,\theta), \quad \mathsf{T}^{\mathrm{FM}}(z,\theta) = \mathrm{tr}_{a,b} \mathsf{M}_{a,b}^{\mathrm{FM}}(z,\theta), \tag{74}$$

such that

$$\left[ \mathsf{T}^{\mathrm{FM}}(z,\theta), \mathsf{T}^{\mathrm{FM}}(w,\theta) \right] = 0, \quad z, w \in \mathbb{C}. \tag{75}$$

Numerically we have checked for systems with $L \leq 14$, where (75) works. Moreover, the Hamiltonian (72) can be obtained as a local conserved charge from the conjectured transfer matrix

$$\mathsf{H}^{\mathrm{FM}} = \mathrm{i} \partial_z \log \mathsf{T}^{\mathrm{FM}}(z,\theta) \big|_{z=1}. \tag{76}$$

This is guaranteed by the conjecture that the R matrix (intertwiner) exists, satisfying the Yang–Baxter relation

$$\mathsf{R}_{(a,b)(c,d)}^{\mathrm{FM}}(z,w,\theta) \mathsf{L}_{a,b,j}^{\mathrm{FM}}(z,\theta) \mathsf{L}_{c,d,j}^{\mathrm{FM}}(w,\theta) = \mathsf{L}_{c,d,j}^{\mathrm{FM}}(w,\theta) \mathsf{L}_{a,b,j}^{\mathrm{FM}}(z,\theta) \mathsf{R}_{(a,b)(c,d)}^{\mathrm{FM}}(z,w,\theta), \tag{77}$$

and

$$\begin{aligned} \mathsf{R}_{(a,b)(c,d)}^{\mathrm{FM}}(z,w,\theta) \mathsf{R}_{(a,b)(e,f)}^{\mathrm{FM}}(z,x,\theta) \mathsf{R}_{(c,d)(e,f)}^{\mathrm{FM}}(w,x,\theta) = \\ \mathsf{R}_{(a,b)(e,f)}^{\mathrm{FM}}(z,x,\theta) \mathsf{R}_{(c,d)(e,f)}^{\mathrm{FM}}(w,x,\theta) \mathsf{R}_{(a,b)(c,d)}^{\mathrm{FM}}(z,w,\theta). \end{aligned} \tag{78}$$

There are a few limits where the R matrix is known, such as $\theta = \frac{n\pi}{2}$, $n \in \mathbb{Z}$, cf. Appendix C. In principle, we would like to obtain the R matrix $\mathsf{R}_{(a,b)(c,d)}^{\mathrm{FM}}(z,w,\theta)$ by requiring it to satisfy the

Yang–Baxter relation (77). However, the procedure is tedious even for symbolic calculation software on a laptop. We shall leave the exact form of the R matrix to later consideration.

Even though it is cumbersome to get the exact expression for the conjectured R matrix, we still can write down two simple limits that the conjectured R matrix has to fulfil,

$$R^{FM}_{(a,b)(c,d)}(z, 1, \theta) = L^{FM}_{a,b,c}(z) L^{FM}_{a,b,d}(z), \tag{79}$$

and

$$R^{FM}_{(a,b)(c,d)}(1, w, \theta) \propto \left( L^{FM}_{c,d,b} \right)^{-1}(w) \left( L^{FM}_{c,d,a} \right)^{-1}(w). \tag{80}$$

These two limits are analytically checked and indeed they are both fulfilled.

In addition to considering the total Hamiltonian (72), there is another generalisation to the Fendley model, similar to the apporach to obtain the $Q$-state chiral Potts model from transverse field Ising model. By doing so, we keep the generalised Onsager algebra GO$(r+1)$ intact. We outline the construction in the following section.

# 5  Generalisation to the chiral-Potts-like cases

Similar to the chiral Potts model as an extension of the TFIM model, while keeping the Onsager algebra intact, we also extend the Fendley model to its chiral counterparts [34]. In order to achieve that, we first generalise the algebra GC$(r, N)$ into GC$(r, N, Q)$, where the generators $h_1, h_2, \cdots, h_N$ satisfy

$$
\begin{aligned}
h_j^Q &= \mathrm{id}, \quad h_j h_{j+m} = \omega h_{j+m} h_j, \quad \omega = \exp\left( \frac{2\mathrm{i}\pi}{Q} \right), \; 1 \le m \le r, \\
h_j h_{j+n} &= h_{j+n} h_j, \quad n \ge r+1, \; 1 \le j \le N,
\end{aligned}
\tag{81}
$$

with the same periodic boundary condition and $Q \in \mathbb{Z}_{>0}$. It is obvious that the algebra GC$(r, N) \equiv$ GC$(r, N, Q = 2)$.

With some algebraic manipulations, we can find presentations of the "coupled graph Temperley-Lieb algebra" and the generalised Onsager algebra GO$(r+1)$ as before. The "coupled graph Temperley-Lieb algebra" is defined as follows:

$$
\begin{aligned}
\left( e_j^{(k)} \right)^2 &= \beta e_j^{(k)}, \quad e_j^{(k)} e_{j\pm m}^{(l)} e_j^{(k)} = e_j^{(k)}, \quad 1 \le m \le r, \; 1 \le k, l \le Q-1, \\
e_j^{(k)} e_{j+n}^{(l)} &= e_{j+n}^{(l)} e_j^{(k)} \quad n \ge r+1, \; 1 \le j \le N, \; 1 \le k, l \le Q-1 \\
e_j^{(k)} e_j^{(l)} &- 0, \quad k \ne l, \; 1 \le k, l \le Q-1.
\end{aligned}
\tag{82}
$$

The algebra GTL$(\beta, r, N)$ is a subalgebra of the "coupled graph Temperley-Lieb algebra", when fixing $e_j \equiv e_j^{(k)}$. When $r = 1$, it becomes the coupled Temperley-Lieb algebra, which has a physically relevant representation being the $Q$-state chiral Potts model [59] with $\beta = \sqrt{N}$.

Here instead we have

$$e_j^{(k)} = \frac{1}{\sqrt{Q}} \sum_{a=1}^{Q} \left( \omega^k h_j \right)^a, \tag{83}$$

with $\omega$ being the $Q$-th root of unity as above, satisfying the relations of the "coupled graph Temperley-Lieb algebra" (82) with $\beta = \sqrt{N}$.

Moreover, for the generalised Onsager algebra GO$(r+1)$, we find the following presentation from GO$(r, N, Q)$:

$$A^{(s)} = \frac{2}{Q} \sum_{j=1}^{N/(r+1)} \sum_{a=1}^{Q-1} \frac{h_{(r+1)j-s}^a}{1 - \omega^{-a}}, \quad s \in \{0, 1, 2, \cdots, r\}, \tag{84}$$

satisfying the Dolan–Grady relation (9) with $N \bmod (r+1) = 0$. When we choose $Q = 2$, we recover the results in the previous sections.

Without delving into the details, we give two representations of the algebras above, one being the $Q$-state chiral Potts model, the other being a chiral-Potts-like generalisation of the Fendley model. The physical properties and other implications are postponed to future investigations.

## 5.1 Representations of $\mathrm{GC}(r, N, Q)$

To begin with, we introduce the clock operators acting on the vector space $(\mathbb{C}^Q)^{\otimes L}$,

$$\mathbf{X}_j = \begin{pmatrix} 0 & 1 & 0 & \cdots & 0 \\ 0 & 0 & 1 & \cdots & 0 \\ \vdots & & \ddots & \ddots & \vdots \\ 0 & 0 & 0 & \cdots & 1 \\ 1 & 0 & 0 & \cdots & 0 \end{pmatrix}_j, \quad \mathbf{Z}_j = \begin{pmatrix} 1 & 0 & \cdots & 0 & 0 \\ 0 & \omega & \cdots & 0 & 0 \\ \vdots & & \ddots & & \vdots \\ 0 & 0 & \cdots & \omega^{Q-2} & 0 \\ 0 & 0 & \cdots & 0 & \omega^{Q-1} \end{pmatrix}_j. \tag{85}$$

The operators satisfy

$$\mathbf{X}_j^Q = \mathbf{Z}_j^Q = \mathbb{1}, \quad \mathbf{X}_j \mathbf{Z}_j = \omega \mathbf{Z}_j \mathbf{X}_j, \quad [\mathbf{X}_j, \mathbf{X}_m] = [\mathbf{Z}_j, \mathbf{Z}_m] = [\mathbf{X}_j, \mathbf{Z}_m] = 0, \ j \neq m. \tag{86}$$

When $Q = 2$, the clock operators become Pauli matrices, $\mathbf{X}_j \to \sigma_j^x$ and $\mathbf{Z}_j \to \sigma_j^z$.

Firstly, we consider the representation of the algebra $\mathrm{GO}(1, 2L, Q)$, where

$$h_{2j-1} \mapsto \mathbf{h}_{2j-1}^{\mathrm{CP}} = \mathbf{Z}_j, \quad h_{2j} \mapsto \mathbf{h}_{2j}^{\mathrm{CP}} \mathbf{X}_j \mathbf{X}_{j+1}^{\dagger}. \tag{87}$$

The chiral Potts model is defined as

$$\mathbf{H}^{\mathrm{CP}} = \sum_{j=1}^{L} \sum_{a=1}^{Q-1} \left( \lambda \frac{\left(\mathbf{h}_{2j-1}^{\mathrm{CP}}\right)^a}{1 - \omega^{-a}} + \frac{\left(\mathbf{h}_{2j}^{\mathrm{CP}}\right)^a}{1 - \omega^{-a}} \right) = \lambda \frac{Q}{2} \mathbf{A}_{\mathrm{CP}}^{(0)} + \frac{Q}{2} \mathbf{A}_{\mathrm{CP}}^{(1)}, \tag{88}$$

with the Onsager generators

$$A^{(0)} \mapsto \mathbf{A}_{\mathrm{CP}}^{(0)} = \frac{2}{Q} \sum_{j=1}^{L} \sum_{a=1}^{Q-1} \frac{\left(\mathbf{h}_{2j-1}^{\mathrm{CP}}\right)^a}{1 - \omega^{-a}}, \quad A^{(1)} \mapsto \mathbf{A}_{\mathrm{CP}}^{(1)} = \frac{2}{Q} \sum_{j=1}^{L} \sum_{a=1}^{Q-1} \frac{\left(\mathbf{h}_{2j}^{\mathrm{CP}}\right)^a}{1 - \omega^{-a}}. \tag{89}$$

It is well-known that the $Q$-state chiral Potts model consists of the two parts forming a representation of the Onsager algebra. The relation to the coupled Temperley-Lieb algebra is similar to the Ising case, and it has been discussed in details in [59].

More interestingly, we can simply generalise the Fendley model in a similar manner. We consider the following representation of the algebra $\mathrm{GC}(r, L, Q)$ acting on the vector space $(\mathbb{C}^Q)^{\otimes L}$,

$$h_j \mapsto \mathbf{h}_j^{\mathrm{CPFM}} = \mathbf{X}_j \mathbf{X}_{j+1} \cdots \mathbf{X}_{j+r-1} \mathbf{Z}_{j+r}. \tag{90}$$

The chiral-Potts-like generalisation of the Fendley model thus can be expressed as

$$\mathbf{H}^{\mathrm{CPFM}} = \sum_{j=1}^{L} \sum_{a=1}^{Q-1} \frac{\left(\mathbf{h}_j^{\mathrm{CPFM}}\right)^a}{1 - \omega^{-a}}, \tag{91}$$

which is Hermitian, the same as the chiral Potts model (88).

**Remark.** If instead we consider another Hamiltonian with open boundary condition,

$$\mathbf{H}' = \sum_{j=1}^{L-r} \xi_j \mathbf{h}_j^{\text{CPFM}}, \quad \xi_j \in \mathbb{R}, \tag{92}$$

we arrive at a non-Hermitian Hamiltonian with free parafermionic [53] spectra, as discussed in details in [35]. In the meantime, the chiral-Potts-like generalisation of the Fendley model (91) is always interacting and Hermitian.

The generalised Onsager algebra $\text{GO}(r + 1)$ has a similar representation with $L \mod (r + 1) = 0$, i.e.

$$A^{(s)} \mapsto \mathbf{A}_{\text{CPFM}}^{(s)} = \frac{2}{Q} \sum_{j=1}^{L/(r+1)} \sum_{a=1}^{Q-1} \frac{\left(\mathbf{h}_{(r+1)j-s}^{\text{CPFM}}\right)^a}{1 - \omega^{-a}}, \tag{93}$$

which leads to the following rewriting of the Hamiltonian (91)

$$\mathbf{H}^{\text{CPFM}} = \sum_{s=0}^{r} \mathbf{A}_{\text{CPFM}}^{(s)}. \tag{94}$$

We expect the phase diagram and the existence of the "dual Hamiltonian" to be the same as the $Q = 2$ case, i.e. the Fendley model discussed previously. However, the phase transitions are expected to be different from the Fendley model. Since the generalised Onsager algebra is present in the generalisation too, we expect that the chiral-Potts-like generalisation is also integrable. Yet the Lax operator and the R matrix are needed to be constructed, which should be more complex than the ones of the Fendley model. We reserve all those intriguing questions for the future work.

## 6 Conclusions and outlooks

In this article, we begin with presenting the generalised Clifford algebra $\text{GC}(r, N)$ and its relation to the graph Temperley-Lieb algebra $\text{GTL}(\beta, r, N)$ and the generalised Onsager algebra $\text{GO}(r + 1)$. Then we continue with its representations. Above all, the Fendley model can be expressed in terms of the operators of the representation of $\text{GC}(r, N)$. This reveals the relation between the Fendley model and the generalised Onsager algebra $\text{GO}(r + 1)$, which has gone undetected previously. The integrability of the Fendley model with the periodic boundary condition is considered next, analogous to the free fermionic eight-vertex model case. We discuss the chiral-Potts-like generalisation of the Fendley model in the end.

The existence of self-dualities is ubiquitous when we consider the generalised Onsager algebra, as demonstrated in previous sections. In fact, with the duality itself, we are able to extract numerous physical properties of the model. This has been well exploited in the "bond algebra approach" [60–63], where the "bond algebra" refers to the algebra of the local terms of the Hamiltonian, i.e. the generalised Clifford algebra $\text{GO}(r, N)$ in the scope of this article. Together with the graph theoretical approach to quantum lattice models [36, 64], it is intriguing to combine different methods to discover new "bond algebras" that possess similar self-duality structures.

Even though we have shown that the Fendley model and its generalisations are closely related to the generalised Onsager algebra, many intriguing aspects have not been discussed in the current article. Most notably, we would like to understand how the generalised Onsager algebra affects the physical properties of the Fendley model and its chiral-Potts-like generalisation, e.g. the phase transitions and its complete spectrum. Meanwhile, it is worth investigating the mathematical structure of the R matrix of the Fendley model and its relation to the

generalised Onsager algebra. The hidden extended supersymmetry algebra shown in [33] is beyond the scope of the article. The relation between the extended supersymmetry algebra and the generalised Onsager algebra is missing at the moment. Ultimately, the generalised Onsager algebra, the quantum integrability and the extended supersymmetry should be comprehended under one roof for the Fendley model and the chiral-Potts-like generalisation. The article should be considered as an appetiser for the more ambitious and intriguing questions both in theoretical physics and mathematics mentioned above.

## Acknowledgements

I would like to thank Alvise Bastianello, Paul Fendley, Vladimir Gritsev, Hosho Katsura, Jules Lamers, Zohar Nussinov, Gerardo Ortiz, Vincent Pasquier, Ana Lucia Retore and Jasper Stokman for helpful discussions. I am grateful to Tamás Gombor and Balázs Pozsgay for suggesting the form of the Lax operator in (73). I acknowledge the support from the GGI BOOST fellowship.

## A  Onsager algebra

The Onsager algebra is an infinite-dimensional Lie algebra. As shown in [10,13], the Dolan–Grady relation for only two generators is isomorphic to the Onsager algebra itself (with infinitely many generators). We summarise the definition of the Onsager algebra as follows.

Consider the generators $\{A_m, G_n | m, n \in \mathbb{Z}\}$, which fulfil the following relations,

$$[A_m, A_n] = 4G_{m-n}, \quad [G_m, A_n] = 2(A_{n+m} - A_{n-m}), \quad [G_m, G_n] = 0.$$  (A.1)

The generators $\{A_m, G_n | m, n \in \mathbb{Z}\}$ form an infinite-dimensional Lie algebra, i.e. the Onsager algebra [19]. There is another isomorphism of the Onsager algebra that is particularly useful when considering certain quantum lattice models, which can be found in [15]. When two of the generators $A^{(0)} := A_0$ and $A^{(1)} := A_1$ satisfy the Dolan-Grady relation (9), we can construst all other generators from (A.1) [10].

## B  The R matrix for the free fermionic eight-vertex models

The R matrix for the Hamiltonian (52) can be written as

$$R_{a,b}^{8V}(z, w, q) = \begin{pmatrix} a_1 & 0 & 0 & d_1 \\ 0 & b_1 & c_1 & 0 \\ 0 & c_2 & b_2 & 0 \\ d_2 & 0 & 0 & a_2 \end{pmatrix}_{a,b}, \qquad (B.1)$$

with $q = \exp i\theta$, such that

$$
\begin{aligned}
a_1 =& a_2 = q^8(z-1)^2(w-1)^2 + 8iq^6(z-w)^2 - 2q^4(z^2(7w^2+2w-1) \\
&+ 2z(w^2+6w+1) - w^2 + 2w + 7) - 8iq^2(z-w)^2 + (z-1)^2(w-1)^2, \\
b_1 =& -b_2 = -4e^{i\pi/4}q(q^2-i)\big[z^2(1+q^4(w-1)-w-2iq^2(w+1)) \\
&-(q^2-i)^2z(w^2-1) + w(1+q^4(w-1)-w+2iq^2(w+1))\big], \\
c_1 =& c_2 = q^8(z-1)^2(w-1)^2 - 8iq^6(z-w)^2 - 2q^4(z^2(7w^2+2w-1) \\
&+ 2z(w^2+6w+1) - w^2 + 2w + 7) + 8iq^2(z-w)^2 + (z-1)^2(w-1)^2, \\
d_1 =& -d_2 = 4e^{-i\pi/4}q(q^2+i)\big[z^2(1+q^4(w-1)-w+2iq^2(w+1)) \\
&-(q^2+i)^2z(w^2-1) + w(1+q^4(w-1)-w-2iq^2(w+1))\big].
\end{aligned}
\tag{B.2}
$$

The R matrix satisfies the Yang–Baxter relation,

$$
R^{8V}_{a,b}(z,w,q)R^{8V}_{a,c}(z,y,q)R^{8V}_{b,c}(w,y,q) = R^{8V}_{b,c}(w,y,q)R^{8V}_{a,c}(z,y,q)R^{8V}_{a,b}(z,w,q).
\tag{B.3}
$$

The R matrix (B.2) with generic values of $q$ seems to be different from the R matrices of free fermionic 8-vertex models in [65, 66]. It would be interesting to understand whether there exists the transformation preserving the Yang–Baxter relations that relates the R matrix (B.2) to the known ones in [66].

## C The R matrix for the Fendley model with $r = 2$

We present the R matrix in (63) for the Fendley model with $r = 2$ explicitly, i.e.

$$
\mathbf{R}^{FM}(u,v) = \begin{pmatrix}
r_1 & 0 & 0 & 0 & 0 & 0 & r_3 & 0 & 0 & 0 & 0 & r_2 & 0 & r_2 & 0 & 0 \\
0 & 0 & r_3 & 0 & r_1 & 0 & 0 & 0 & 0 & r_2 & 0 & 0 & 0 & 0 & 0 & r_2 \\
0 & 0 & 0 & r_2 & 0 & r_2 & 0 & 0 & r_1 & 0 & 0 & 0 & 0 & 0 & r_3 & 0 \\
0 & r_2 & 0 & 0 & 0 & 0 & 0 & r_2 & 0 & 0 & r_3 & 0 & r_1 & 0 & 0 & 0 \\
0 & r_1 & 0 & 0 & 0 & 0 & 0 & -r_3 & 0 & 0 & r_2 & 0 & -r_2 & 0 & 0 & 0 \\
0 & 0 & 0 & -r_3 & 0 & 1 & 0 & 0 & -r_2 & 0 & 0 & 0 & 0 & 0 & r_2 & 0 \\
0 & 0 & r_2 & 0 & -r_2 & 0 & 0 & 0 & 0 & r_1 & 0 & 0 & 0 & 0 & 0 & -r_3 \\
-r_2 & 0 & 0 & 0 & 0 & 0 & r_2 & 0 & 0 & 0 & 0 & -r_3 & 0 & r_1 & 0 & 0 \\
0 & 0 & r_1 & 0 & -r_3 & 0 & 0 & 0 & 0 & -r_2 & 0 & 0 & 0 & 0 & 0 & r_2 \\
-r_3 & 0 & 0 & 0 & 0 & 0 & r_1 & 0 & 0 & 0 & 0 & r_2 & 0 & -r_2 & 0 & 0 \\
0 & -r_2 & 0 & 0 & 0 & 0 & 0 & r_2 & 0 & 0 & r_1 & 0 & -r_3 & 0 & 0 & 0 \\
0 & 0 & 0 & r_2 & 0 & -r_2 & 0 & 0 & -r_3 & 0 & 0 & 0 & 0 & 0 & r_1 & 0 \\
0 & 0 & r_1 & 0 & r_3 & 0 & 0 & 0 & 0 & -r_2 & 0 & 0 & 0 & 0 & 0 & -r_2 \\
0 & r_3 & 0 & 0 & 0 & 0 & 0 & r_1 & 0 & 0 & -r_2 & 0 & -r_2 & 0 & 0 & 0 \\
-r_2 & 0 & 0 & 0 & 0 & 0 & -r_2 & 0 & 0 & 0 & 0 & r_1 & 0 & r_3 & 0 & 0 \\
0 & 0 & -r_2 & 0 & -r_2 & 0 & 0 & 0 & 0 & r_3 & 0 & 0 & 0 & 0 & 0 & r_1
\end{pmatrix},
\tag{C.1}
$$

where

$$
r_1 = 1, \quad r_2 = \tanh(u-v), \quad r_3 = -\tanh(u-v)\tanh(u+v).
\tag{C.2}
$$

It also satisfies the following Yang–Baxter relation,

$$
\begin{aligned}
\mathbf{R}^{FM}_{(a,b)(c,d)}(z,w,\theta)&\mathbf{R}^{FM}_{(a,b)(e,f)}(z,x,\theta)\mathbf{R}^{FM}_{(c,d)(e,f)}(w,x,\theta) = \\
\mathbf{R}^{FM}_{(a,b)(e,f)}(z,x,\theta)&\mathbf{R}^{FM}_{(c,d)(e,f)}(w,x,\theta)\mathbf{R}^{FM}_{(a,b)(c,d)}(z,w,\theta).
\end{aligned}
\tag{C.3}
$$

The R matrix for the Fendley model with $r = 2$ depends on both $(u-v)$ and $(u+v)$, hence it cannot be brought into a different form, differnt from the case of the free fermionic eight-vertex model (50).

When $v = 0$, we have

$$\mathbf{R}^{\mathrm{FM}}_{(a,b)(c,d)}(u,0) = \mathbf{L}^{\mathrm{FM}}_{a,b,c}(u)\mathbf{L}^{\mathrm{FM}}_{a,b,d}(u) = (\mathbb{1} + u\mathbf{h}^{\mathrm{FM}}_{a,b,c})\mathbf{P}_{b,c}\mathbf{P}_{a,c}. \tag{C.4}$$

Moreover, the inverse of the R matrix can be obtained by permuting the auxiliary spaces and spectral parameters, i.e.

$$\mathbf{R}^{\mathrm{FM}}_{(a,b)(c,d)}(u,v)\mathbf{R}^{\mathrm{FM}}_{(c,d)(a,b)}(v,u) = \frac{2[\cosh(4u) + \cosh(4v) - 2\sinh(2u)\sinh(2v)]}{[\cosh(2u) + \cosh(2v)]^2}. \tag{C.5}$$

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
