# Peer review of "Generalised Onsager Algebra in Quantum Lattice Models"

_SciPost Physics, doi:SciPost Phys. 13, 070 (2022)_

## Round 2 · Referee Report · Anonymous (Referee 1) · 2022-7-24

Strengths

  1. new integrable models related with generalized Onsager algebras
  2. new examples of Yang-Baxter algebras

Weaknesses

see report

Report

A. Overview

The paper contains interesting results about generalizations of the Onsager algebra, Yang-Baxter algebras and lattice models recently introduced and studied by Fendley, Alcaraz-Pimenta, Minami, Miao,.... The paper also opens interesting perspectives. Before publication in SciPost, the content should be however revised according to the comments below.

B. Details

  1. About the quotient of GTL$(\sqrt(2),r,N)$ mentionned around (2.5). This part should be improved.

a. The author is considering a quotient of GTL$(\sqrt(2),r,N)$, by some relations (for instance (2.6)). Let's denote this quotient by GTL' and its generators by $e'_i$. Instead of (2.5) a map $GTL' \rightarrow GC(r,N)$ should be defined (equality in (2.5) doesn't hold): $e'_j \rightarrow 1/\sqrt(2)(id + h_j)$

b. The author says that additionnal relations such as (2.6) hold in GTL' compared with GTL. Is it possible to characterize the quotient by identifying all additional relations? A comment is welcome around (2.6).

  1. Below (2.6) and top of page 5, a more precise reference where GO(r+1) is welcome (Definition number? Theorem number?)

  2. In (2.11), (2.12), equalities should not be used since the maps are not isomorphisms. Arrows should be used. For instance a map $GO(r+1) \rightarrow GC(r,N)$ is given by $A^{(s)}\rightarrow ...$ instead of (2.11). Similar for (2.12). The conclusion below (2.12) should be improved accordingly.

  3. About (2.13)-(2.15). The author explains that (2.14) holds thanks to (2.1). Details of the proof are not given, but I understand it is straightforward. A corollary is that (2.15) commutes with $A_0,A_1$. Then it is said 'Details are demonstrated in Appendix A'. This sentence is not suitable, as details are not given. In Appendix A, it is only recalled that the Onsager algebra admits two presentations, either (A.1) or the one with generators $A_0,A_1$ satisfying (2.9). So, any operator commuting with $A_0,A_1$, commutes with $A_n,G_m$. So, the sentence 'Details...Appendix A' should be modified.

  4. Begining of section 3, it would be helpful to add a sentence explaining why reviewing known cases is useful and/or necessary for the rest of the paper.

  5. Section 3.1. The sentence 'we make the following homomorphism to a vector space...' has to be improved. I guess the author means that he gives a homomorphism from $GC(1,2L)$ to an algebra of matrices in $End((C^2)^{\otimes L})$. It is simpler to write above (3.1):

In this case, we have the map: $GC(1,2L) \rightarrow End({C^2}^{\otimes L})$, then giving (3.1)

  1. Above (3.2), it is written $GC(2,2L)$. Is this not a misprint? $GC(2,2L)\rightarrow GC(1,2L)$?

  2. Below (3.6). The specific terminology 'spin current' is chosen. Please explain.

  3. Below (3.6) 'presence of Onsager algebra' $\rightarrow$ 'presence of Onsager algebra's symmetry'.

  4. About (3.7)-(3.9) and footnote 2. The presentation should be improved, simply giving the homomorphism (by analogy with section 3.1)

$ GC(1,L)\rightarrow End((C^2)^{\otimes L})$

Footnote 2 is then useless.

  1. Top of page 8: constructing $\rightarrow$ studying.

  2. Eq (3.11): is it not a misprint $L \rightarrow L/2$ in the first sum of $A_8V^(1)$?

  3. Around equation (3.15), the author claims that Fendley's model with periodic boundary conditions is interacting, contrary to the case with free boundary conditions, which is free-fermionic. This seems to be a surprising result. In fact, the case $r=1$ with p.b.c. (Ising chain) is also free-fermionic. The author should clarify what he means by interacting' model, and at least to present some numerical evidence that this is indeed the case.

  4. Around equation (3.16), the author seems to imply that the Hamiltonian (3.15) is only integrable for homogeneous couplings. The model should be integrable for any couplings according to Ref. [27]. Please verify.

  5. In the sentence before (3.23), it seems that one should have $GTL(L,r)$ instead of $GTC(L,r)$.

  6. Around equation (3.26), could the author expand on the Kramers-Wannier duality? What is a `generalization of the phase transition in the TIFM'?

  7. In Section 3.3., same remark as above. Please indicate the homomorphism precisely: map $GC(r,L) \rightarrow End((C^2)^{\otimes L})$.

  8. In begining of section 4, it is written 'As we have shown in the previous sections...'. However, saying 'As we have recalled in the previous sections..' is more appropriate, since it seems all results are taken from the literature. If new results are proven, it should be pointed out which ones.

  9. In the third line of begining of section 4, it should be written $GL(r,L), GL(L,r)$ (because of duality) and $GO(r+1)$.

  10. Last sentence of 1st paragraph of begining of Section 4: 'generic case' $\rightarrow$ 'generic integer values of r'

  11. The procedure of building a R-matrix from a Temperley-Lieb type generator is generally known as ``Baxterization''. Maybe the author can add this information and the reference {\it V. F. R. Jones, “Baxterization,” Int. J. Mod. Phys. A6 (1991) 2035-2043} around eq. (4.3).

  12. The R-matrix (B.1) is intriguing. Does it fit into previous classifications of 8-vertex type R-matrices? See for example arXiv:1311.4994, arXiv:2010.11231 and references therein.

  13. It seems that the algebra (5.1) was introduced in [28]. If so, the reference should appear.

  14. Correct misprints in eqs. (5.8) and (5.9).

  15. About Introduction: The author gives a review on the subject of Onsager algebras and extensions, in relation with Yang-Baxter algebras. For clarity of the exposal of the subject, the introduction should be improved taking into account of the following informations.

a. About the Onsager algebra (a fixed point subalgebra of affine $sl(2)$). The main topic of the paper is the q=1 case (Onsager algebra), so it is natural to add the q=1 version of the presentation of the Onsager algebra studied in [23] (called alternating, different from the original one [5]). It has been introduced in arXiv:1806.07232, and its central extension has been studied in arXiv:2104.08106. The author also cites important references on the Onsager algebra and recall how it relates with some Yang-Baxter algebras. Presentations of Onsager algebras using Yang-Baxter algebras have previously appear in the literature, and that should be mentionned. See arxiv:1709.08555 for the Onsager algebra using non-standard classical Yang-Baxter algebras.

b. About the q-extensions of the Onsager algebra (a coideal subalgebra of affine $U_q(\widehat{sl(2)}$). The q-deformed extension of the Onsager algebra in its original presentation [5] is introduced and studied in arxiv:1706.08747. The q-deformed extension of the Onsager algebra in its alternating presentation [arXiv:1806.07232, arXiv:2104.08106] is introduced in arXiv:0906.1482. In [23], proofs of existing conjectures in the literature are given.

c. About generalized q-Onsager algebras As can be checked from the introduction of [26], the generalized q-Onsager algebras were first introduced in arXiv:0906.1215.

d. About generalized Onsager algebras. The paper is about generalized Onsager algebras and Yang-Baxter algebras. It is worth pointing out that the same year as [1], the Yang-Baxter algebra presentation of the generalized Onsager algebras was given in arxiv:1709.08555. For the $sl(N)$ case, it is discussed in details in arXiv:1811.02763.

Requested changes

see report

---

## Round 3 · Referee Report · Anonymous (Referee 1) · 2022-8-15

Report

I thank the author for the revised version. Most of the points addressed in the previous report have been answered properly or corrected. However, three points remain to be completed or clarified, and a typo should be corrected.

a. Below (2.6), a comment about the possible existence of additionnal relations should be given. Is the quotient characterized only by (2.3),(2.6) or additionnal relations may appear? If the author doesn't know, that should be said explicitly.

b. About 8 of the first report. The answer is not satisfying in the present revised form. Indeed, to support the hypothesis that Fendley's model for periodic boundary conditions (pbc) is not interacting (at least for small sizes), let us observe the following. It is correct to say that Fendley's method only provides the quasi-energies and raising/lowering operators for the open boundary case. However, this does not exclude the possibility of finding them using some other way (yet to be found) for the pbc case. In fact, for small lattices one can indeed find the quasi-energies by brute force. For the case depicted in Fig.2, let e_1=\sqrt{3},e_2=1+\sqrt{2},e_3=1-\sqrt{2}. One can find by brute force that the eigenvalues of Fig.2 are given by,

e_1+e_1

-e_1-e_1

e_1-e_1

e_2 + e_3

e_2 - e_3

  • e_2 + e_3

  • e_2 - e_3

This case may look trivial but larger chains could be analyzed by brute force as well. Note that the above structure is also present in periodic Ising. Indeed, the definition (1) in ref [53] does not really apply to the Ising case with pbc - this is related to the fermionic parity sectors. So, maybe Fendley's model with pbc is similar to Ising with pbc.

So, if the author asserts that Fendley's model for pbc is interacting, that should be clearly proven. I can not find such statement in Fendley's paper. So, the author should indicate the precise location in the text of Fendley's paper where such statement is justified, or give a precise reference in the literature, or give a proof of it (contradicting the above examples for small sizes). Otherwize, the claim that Fendley's model for pbc is interacting should be removed from the text. Saying that it is hard to solve it for pbc is a different issue.

c. About point 9 of the first report. The answer is not satisfying in the present revised form. Indeed, while the transfer matrix approach for the open case does not apply to periodic pbc, in Appendix A of ref. [33] a different approach is provided, and it is also suitable for inhomogeneous couplings (as explained in the last paragraph of Appendix A of [33]). So, whereas it is correct to say that the approach proposed by the author (different from [33]) exhibits integrability, it doesn't exclude the possibility that the model is integrable for inhomogeneous couplings. Being integrable or admitting a Lax operator construction are not equivalent statements. The second statement implies the first, but the reverse is not true in general.

d. Typo middle page 2 in introduction: the q-Onsager algebra [?, 24, 25]

Provided the above changes, I think the new revised version is suitable for publication in SciPost.

Requested changes

See report

---

## Round 3 · Author Response

Dear referee and editor,

I am grateful for the referee for his valuable comments and suggestions on the draft. I have improved the draft according to the referee's suggestions. The list of changes are given below

---

## Round 3 · List of Changes

1. I have improved the part about the quotient of $\mathrm{GTL} (\sqrt{2}, r , N)$ around (2.5). The map (2.5) has been stated more clearly now.

  2. Below (2.6) I have given the precise reference of the Definition in [1].

  3. I have corrected the maps in (2.11) and (2.12) according to the referee's suggestion.

  4. The part about (2.15) commutes with $\mathbf{A}_n$ has been rewritten.

  5. I added a sentence to explain why we review the known cases in the first paragraph of Sec. 3.

  6. I have improved the language used near (3.1), (3.2), (3.7) and Sec. 3.3 according to the referee's suggestion. I deleted the previous footnote near (3.7).

  7. Below (3.6), I have explained the origin of the terminology "spin current" as the spin current in spin-1/2 XXZ model.

  8. About 13. in the referee's report, I added a figure (Fig. 2 in the updated version) and explained the reason why Fendley models with periodic boundary should be interacting; namely, the non-local transformation in Fendley '19 does not apply any more. But Jordan-Wigner transformation (for TFIM) still works with periodic boundary. This has been observed in Fendley '19 already, and stated explicitly in its abstract.

  9. About 14. in the referee's report, in fact in Fendley '19, the author constructed extensively-many conserved quantities for the Fendley model with open boundary condition and inhomogenous couplings. This will not work for the periodic boundary condition. In the periodic case, I used the Lax operator to obtain transfer matrix. In principle, I could put inhomogeneities in the Lax operator and get a model that is still integrable. However, it is not the same Fendley model with inhomogeneous couplings, but a model with local density of longer-range terms. That is not what Fendley '19 studied. In short, the statement about Fendley model with periodic boundary and inhomogeneous couplings is not integrable stays correct.

  10. I changed the word "the generalisation of the phase transition in the TFIM" into "analogous to the phase transition in the TFIM".

  11. About 22. in the referee's report, I have checked briefly and the R matrix (B.1) does not fit the known categories in the two references. I have made a short comment in the updated version.

  12. I have added the references recommendated by the referee. Mainly in the introduction and the part related to the baxterisation.

  13. I have changed typos mentioned in the referee's report (9., 11., 12., 15., 18., 19. 20., 23. and 24. there).

---

## Round 4 · Referee Report · Anonymous (Referee 2) · 2022-8-20

Strengths

This is a well written, very useful paper where the authors clarify relations between various fancy algebras related with integrable models - Temperley-Lieb, Onsager - and further, point out relationships with recent models of interest such as the ones proposed by P. Fendley. In what is usually a rather technical and somewhat unnecessarily mathematical area, I found the paper refreshing, and I am sure it will become a standard reference.

Weaknesses

There were points that deserved improvement, but these were taken take of thanks to the work of the first referee - sorry for contributing my own report a little late.

Report

I think the paper is now ready for publication.

---

## Round 4 · Referee Report · Anonymous (Referee 1) · 2022-8-20

Report

I thank the author for clarifications and changes. In my view, the revised version of the paper is suitable for publication in SciPost.

---

## Round 4 · Author Response

Dear referee and editor,

I am grateful for the referee for his valuable comments and suggestions on the draft. I have improved the draft according to the referee's suggestions. The list of changes are given below:

---

## Round 4 · List of Changes

1. I have improved the text about whether the Fendley model with periodic boundary condition is interacting. The improved part in page 9 is
"
However, the spectra of the Hamiltonians do not satisfy the free fermionic condition in [53] with the periodic boundary condition. This can be observed from numerically obtaining the eigenvalues of (3.14), as shown in Fig. 2. In principle, this do not exclude the possibility that the spectra cannot be partitioned into subsectors that are free fermionic. The most notable example is the TFIM with periodic boundary condition, where the spectrum can be divided into two parts that are free fermionic. In the case of the Fendley model, it is less clear whether such partition of the spectrum into free fermionic parts exists. Moreover, unlike the TFIM, the non-local transformation constructed in [33] no longer applies for periodic boundary. The question whether the Fendley model is intrinsically interacting is postponed to future investigation.
"
Upon the comments of the referee, it is difficult to determine whether the model is interacting or the spectra could be partitioned into parts that are free fermionic. The referee commented that maybe one can divide the spectra into parts that are free. Indeed, it seems possible. But with different degeneracies, it is hard to construct a systematic way of doing so, in my opinion. This seems to be different from transverse field Ising model.

2. I agree with the referee's comment and I added the following remark in page 9:
"
Remark. The transfer matrix in [33] can be applied to the $r = 2$ Fendley model with inhomogeneous couplings and periodic boundary. However, the method in Sec. 4.2 only works for the homogenous case (3.16). Instead, we can add inhomogeneities in the transfer matrix (4.22) constructed in Sec. 4.2, which will result in another Hamiltonian with longer-range interaction.
"

3. The broken reference link in page 2 is fixed.

4. I corrected a typo in Eq. (B.2) and a wrong statement that Ref. [66] can only be applied to R matrices of difference form.

---

## Editorial Decision

published